# Analysis of Machine Learning Classification Approaches for Predicting Students' Programming Aptitude

**Ali Çetinkaya [1,*]**, **Ömer Kaan Baykan [1]** and **Havva Kırgız [2]**

1 Department of Computer Engineering, Konya Technical University, Konya 42250, Türkiye; okbaykan@ktun.edu.tr
2 Konya Science Center, Konya 42100, Türkiye; kirgizhavva@gmail.com
* Correspondence: ali.cetinkaya@gmail.com

**Abstract:** With the increasing prevalence and significance of computer programming, a crucial challenge that lies ahead of teachers and parents is to identify students adept at computer programming and direct them to relevant programming fields. As most studies on students' coding abilities focus on elementary, high school, and university students in developed countries, we aimed to determine the coding abilities of middle school students in Turkey. We first administered a three-part spatial test to 600 secondary school students, of whom 400 completed the survey and the 20-level Classic Maze course on Code.org. We then employed four machine learning (ML) algorithms, namely, support vector machine (SVM), decision tree, k-nearest neighbor, and quadratic discriminant to classify the coding abilities of these students using spatial test and Code.org platform data. SVM yielded the most accurate results and can thus be considered a suitable ML technique to determine the coding abilities of participants. This article promotes quality education and coding skills for workforce development and sustainable industrialization, aligned with the United Nations Sustainable Development Goals.

**Keywords:** machine learning; classification; Code.org; middle school students; coding abilities

## 1. Introduction

In today's rapidly evolving digitalized world, each student should have the opportunity to learn about computer algorithms, how to create apps, and how the internet functions. Therefore, teaching students programming at a young age is essential. As vital as it is for kids to grasp how to solve a given equation or how plants function, it is equally crucial that they comprehend what a "for loop" is, how it is utilized, and how to develop algorithms. The landscape of the labor market is quickly changing owing to the unparalleled rate of technological advancements. Thus, it is imperative for individuals who aspire to be successful in acquiring new skills through lifelong learning. A workforce that is highly data-driven and uses technology is necessary to introduce fresh concepts in the market [1]. Given the swift technological changes brought about by AI, quality education and coding skills for workforce development aligned with the United Nations Sustainable Development Goals will become crucial for sustained economic growth and global competitiveness. Thus, it is vital to use AI and big data to assess middle school students' aptitudes in these areas early on.

Data and analytics are core components of organizations operating in various fields. Therefore, the sooner students are introduced to the coding and AI-related curriculum [2], the sooner they will be able to benefit from employment in these institutions. However, while 90% of parents demand computer lessons in high schools, 53% of schools offer computer-based instruction [1]. A major challenge faced by teachers and parents is to identify students adept at programming and to guide them to programming areas. Teachers have limited time and resources to determine whether students possess coding skills effectively. These resources should be directed at the students who need them most.

However, it is likely that a very talented student body has not had the opportunity to learn coding.

Most studies concerning students' coding skills focus on elementary, high school, and university students. There are only a few studies on predicting middle school students' programming skills, and thus, there is still a paucity of information about coding abilities of middle school students. Refs. [3,4] reported that secondary school students put markedly more effort into certain basic concepts in the context of programming than other students.

While the Scratch app is widely used to teach coding, the Code.org platform brings students and educators worldwide together. This platform provides the opportunity to attend one-hour coding lessons in over 200 different events in more than 180 countries [1]. Over 100 million students around the world have had the opportunity to experience the Hour of Code. Code.org is an online visual platform that allows students to learn to collaborate, develop problem-solving skills, and create computer programs to help them complete difficult tasks. At the end of a course on Code.org, students create their own custom game or story that they can share with other students [1].

Students need to have the capacity to comprehend, analyze, and reason correctly in order to develop their programming abilities and perform a specific programming task. Since the 1960s, spatial tests have been used in programming to measure this aptitude. The most popular of these is the Programmer Aptitude Test (PAT) conducted by IBM [5]. Scholars have recently found that students' spatial skills are highly correlated with their programming skills [6]. The use of spatial skills in introductory computer courses has been demonstrated to enhance students' spatial skills and programming skills [7]. The individual difference related to programming success identified in the literature is the spatial ability. Discussions with University of Nottingham alumni revealed that it is increasingly necessary for them to pass spatial tests as part of the hiring procedure for a programming career. Spatial skill is a cognitive feature that gives a measure of the ability to conceptualize spatial connections between objects [8]. It is widely accepted as an indicator of success in completing computer programming tasks [9].

Machine learning (ML) is a subfield of artificial intelligence (AI) and computer science that concentrates on the use of data and algorithms to simulate the way people learn while continually enhancing the accuracy of these algorithms [10,11]. In applications such as pharmaceuticals, education, e-mail filtering, speech recognition, and image processing, ML techniques are applied when it is challenging or impossible to create standard algorithms to execute required tasks [12].

The use of ML in predicting student success has received growing attention. In ML, prediction refers to the output of an algorithm trained on a historical dataset, evaluating the likelihood of a specific result, and applying it to fresh data [13]. In this context, numerous studies have been carried out on subjects such as estimating 'students' academic success and grades [14–18]. Commonly used algorithms for predicting students' academic achievement and grades include support vector machines (SVM), decision tree (DT), k-nearest neighbor (KNN), random forest (RF), and naive Bayes (NB), among others.

### 1.1. Research Purpose

In this study, we utilized a variety of widely known classification techniques for ML in the field of AI. Our purpose was to compare the effectiveness of the algorithms and determine which approach is suitable for classifying the programming ability of students.

### 1.2. Research Questions

This research examined the application of machine learning (ML) algorithms for categorizing the coding abilities of middle school students. We sought to answer the following research questions:

RQ1: Can middle school students' coding abilities be assessed using a spatial test?

RQ2: What are the most effective ML classification approaches for classifying student aptitude?

The remainder of this paper is structured as follows. Section 2 presents an overview of the extant literature. Section 3 discusses the materials and methods. Section 4 provides the experimental results and discussions, and Section 5 concludes and summarizes this study.

## 2. Related Work

The literature review relevant to this article is organized under three subheadings: machine learning, spatial skills, and block-based programming.

### 2.1. Machine Learning

Educational data mining refers to the methodologies, tools, and investigations pertaining to mining information from the massive amounts of data produced by student-related studies in educational institutions [19].

Various studies have been conducted in the realm of education. Kovacic [20] utilized DT models and attained a proper classification rate of 60.5% overall to pre-identify successful and unsuccessful students. In addition, the feature importance technique was applied to determine the crucial background characteristics such as academic and socio-demographic variables. Yadav et al. [21] assessed the validity of prediction models provided by ML techniques for supervising pupil retention. Compared to other ML techniques, DT was found to provide classification rules that are more straightforward.

To forecast the academic performance of engineering students, researchers propounded a DT-based multilevel classification model. At Level 1 of this approach, the researchers focused on the development, assessment, and comparison of four distinct classification models [22]. In Level 2, they focused on improving the performance of individual classifiers.

Morais et al. [23] investigated the data of 262 students enrolled in remote education and 161 students enrolled on campus. The researchers identified SVM as the most effective classification method for both sets of data. Kolo and Adepoju [24], employed various classification algorithms to forecast the academic performance of students and sought the most effective algorithms among them. Compared with Rep-tree, SimpleCart, decision table, and J48 ML algorithms, neural network (NN)-based classification were found to be the most accurate, followed by NB and ID3 methods.

Sikder et al. [25] predicted students' annual CGPA using NN and found that the projected values corresponded to the actual CGPA values. Saa proposed a categorization model for predicting students' performance by taking into consideration their personal, social, and academic data. They identified intriguing patterns using NB and DT. Of the four DT algorithms that they employed, namely, C4.5, ID3, CART, and CHAID, they found CART to be the most accurate [26]. Hsieh et al. [27] used the Jacobian matrix-based learning machine to evaluate students' learning performance. Han et al. [28] applied the AdaBoost assembly algorithm to predict student classification and demonstrated that the algorithm showed superior performance compared with techniques such as DT, artificial neural network (ANN), and SVM.

Tampakas et al. [29] employed two-level data mining methods to identify students at risk of not completing their studies and their graduation time. Hussain et al. [30] evaluated 24 socioeconomic, demographic, and academic characteristics of 300 students from three different universities. They employed four classification algorithms to predict student performance. The RF method was found to have an accuracy of 99.9%, the greatest accuracy of all algorithms.

Miguéis et al. [31] used a data set of 2459 students, spanning the years from 2003 to 2015, from a European Engineering School of a public research University to demonstrate the ability of the proposed classification model to predict the students' performance level with an accuracy above 95%, in an early stage of the students' academic path. They found that random forests are superior to the other classification techniques that were considered (decision trees, support vector machines, naive Bayes, bagged trees and boosted trees).

Salal et al. [32] predicted the academic achievement of secondary school students by classifying 649 individuals based on 33 characteristics. These characteristics were associated

with a student's academic performance, demography, social standing, and schools. For performance prediction, they employed several classification algorithms, including DT (J48), NB, RF, REPTree, random tree, JRip, OneR, and ZeroR.

Berens et al. [33] introduced an approach that integrates logit regression modeling, neural network modeling, and decision trees to forecast the likely dropout of academic students.

Liao et al. [34] presented a method that uses early-term clicker data and a support vector machine to predict student final exam scores, allowing early identification and assistance for struggling students across various courses and institutions.

Chen et al. [35,36] gave a broad overview of the literature on artificial intelligence and machine learning in the education field. Hwang et al. [37] proposed a framework to help researchers with computer and educational backgrounds to identify considerations for implementing machine learning in different educational institutions. Chen et al. [38] identified gaps in both the application and theory of educational artificial intelligence.

Rastrollo-Guerrero et al. [39] evaluated over 70 publications to demonstrate the many current methodologies extensively used for forecasting student performance as well as the goals they must accomplish in predict students' performance. The studied AI-based techniques and methodologies include, among others, ML, collaborative filtering, recommender systems, and ANN.

Akmeşe et al. [40] employed an ML-based prediction method in lieu of conventional graphics and descriptive statistics. To evaluate the academic achievement of pupils, they studied demographic and socioeconomic factors such as age, gender, city, family income, and family education level. Pallathadka et al. [41] explored ML algorithms such as NB, ID3, C4.5, and SVM using the UCI machinery student performance dataset. They assessed the algorithms using factors such as precision and error rate. Siddique et al. [42] observed three single classifiers, namely, multilayer perceptron (MLP), J48, and PART, in addition to three well-established ensemble techniques, namely bagging (BAG), multiboost (MB), and voting (VT). In their experiments, MB with MLP surpassed the competition in the evaluation, attaining 98.7% accuracy, 98.6% precision, recall, and F-score. The researchers deduced that their proposed approach might be effective for identifying the academic performance of secondary pupils at an early stage to enhance learning outcomes.

Bognar and Fauszt [43] predicted the academic success of university students, assessed the skills of the parties involved in machine learning, and evaluated the key factors and circumstances affecting the reliability of the forecasts.

Bacci and Bertaccini [44] proposed a mixture of hidden Markov model to classify students into groups that are homogenous in terms of university paths, with the aim of detecting bottlenecks in the academic career and improving students' performance.

Using academic and demographic data, Alboaneen et al. [45] created a web-based method for forecasting academic success and identifying students at risk of failing. The ML model developed to forecast the final grade of a course early on. Support Vector Machine (SVM), Random Forest (RF), K-Nearest Neighbors (KNN), Artificial Neural Network (ANN), and Linear Regression (LR) are among the machine learning methods that are used.

### 2.2. Spatial Skills

In recent years, the scientific community in computer science education has begun to demonstrate the relationship between spatial skills and computer science. Research has proven that the average skill grows with academic development [46]. Spatial abilities entail the internal consolidation and manipulation of structures and processes, often in regard to space and form. It is an umbrella term that includes a wide range of skills associated with spatial understanding, such as mental rotation, spatial relations, and closure speed [47].

Studies have revealed a correlation between spatial skills and certain computing skills, such as expression assessment [48], source code navigation [9], complicated exam questions [49], and standardized computer exams [7,50]. It has been established that spatial skills training improves computing outcomes [7,50].

### 2.3. Blocked-Based Programming

Providing block-based learning environments with visual aids and engaging additional sense organs is known to make programming enjoyable. Delivering learning material in the forms of audio, text, visual, animation, and simulation is likely to capture the interest of students. In these contexts, the programs are shown as blocks, and the drag-and-drop approach is utilized. This prevents users from forgetting their codes and makes it easier for them to translate their thoughts into code [51,52].

The Code.org platform consists of a hierarchical information structure that permits the adoption of a single command through a succession of programming themes. The material is arranged in the form of lessons, but the context of learning is presented in the form of a narrative [53], which can be engaging, especially for younger children. Furthermore, Code.org classes are tailored for children of all ages. Considering the aforementioned, the Code.org platform is especially beneficial for coding programs that target younger children [54].

In this study, we used the spatial test and Code.org levels to categorize the coding abilities of middle school students, thus trying to fill the previously mentioned gap in the literature.

## 3. Materials and Methods

The Materials and methods section is organized under four subsections: data collection, spatial test, Code.org, and methodology.

### 3.1. Data Collection

This study was conducted with 600 students from a secondary school in Konya. Students are enrolled in a public school, which does not administer proficiency exams to determine enrollment. The socioeconomic status of the majority of the students' families falls within the lower- and middle-income brackets.

The students were first administered a four-part spatial reasoning test. Based on their responses, 400 students (204 female and 196 male) were selected to participate in the Classic Maze Course (CMC), which consists of 20 levels. The course was facilitated through the online platform Code.org, indicating that the students accessed and completed the course digitally. We used four ML techniques, namely, SVM, DT, KNN, and quadratic discriminant (QD), on the spatial test and Code.org platform data to classify the coding skills of these students. We further compared the performance of the applied ML approaches.

### 3.2. Spatial Test

To develop excellent programming abilities, students must have the capacity to grasp, analyze, and reason properly in order to perform a specific programming assignment. Programming is not a theoretical topic, but it requires students to be able to reason logically and think critically to complete the given assignment.

Numerous studies, as well as computer science classes, have attempted to identify particular testable characteristics that have a strong correlation with achievement in computer science [55].

Programming aptitude tests have been used in careers and education in the programming sector for several decades. PAT, administered by IBM, is possibly the most popular early exam for assessing programming aptitude [5]. Researchers have examined the connection between programming talent development and several individual characteristics. Programming skill acquisition and other measures of individual differences have been the subject of several studies on, for example, background in mathematics, problem-solving ability, and cognitive skills [56,57].

In this study, we developed a spatial test consisting of three parts: The first part enquired about participants' electronic device ownership and usage. The second part was the paper folding section compiled using various paper folding questions [58]. The participants were assigned three paper folding tasks of varied difficulty (Figure 1). They

were tasked with discerning the appearance of the openings on the unfolded face of the folded paper.

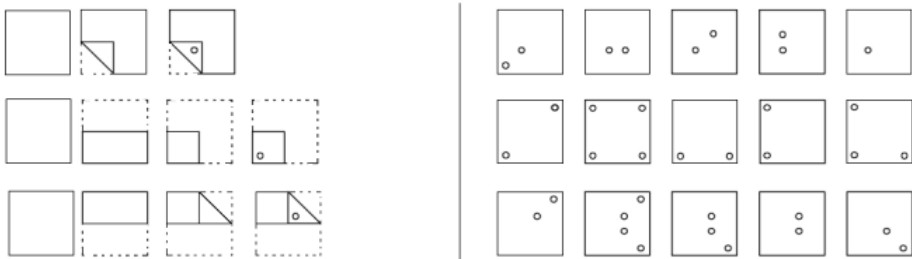

**Figure 1.** Paper folding questions compiled from [58].

The third section was the critical thinking section, which was intended to include numerous sites, including airports, parklands, schools, lakes, hospitals, libraries, and science centers. Participants were instructed to select two locations they would want to visit. The participants then picked two numbers from 1 to 7 that corresponded to distinct routes on the map between the starting and ending places. After selecting the road number, players had to go from the starting point to the first stop, then to the second stop, and finally to the finish line [46]. The part on analytical reasoning had two fundamental questions derived from the Turkish Ministry of National Education's Department of Assessment, Evaluation, and Examination Services' general aptitude tests.

Each ML model was fed with seven input parameters. The first two parameters stemmed from the frequency of participants' usage or ownership of electronic devices like smartphones, tablets, and laptops. The next three parameters were based on participants' performance on paper folding problems to assess spatial reasoning. The last two parameters were derived from participants' answers to analytical reasoning questions such as problem-solving, critical thinking, and decision-making. The responses to all these questions were encoded into numerical values, factoring in the difficulty level rated by experts. These parameters, reflecting aspects like device usage, spatial reasoning, and analytical problem-solving, were then used to train the ML models to classify participants' coding ability.

*3.3. Code.org*

Code.org is a nonprofit organization with the objective of enabling every student to acquire computer science skills similar to conventional sciences [1] without charging any fee. Code.org is a web-based programming environment for children in kindergarten through eighth grade accessible online from any computer or mobile device. In addition, Code.org has a website that leverages the block-based programming paradigm and offers courses in visual programming for computer science education. Training in this environment based on the logic of visual programming gives students and teachers flexibility [59].

Code.org provides a plethora of resources for computer science education. The learning material is presented in the form of lectures and courses, and programming topics are illustrated using diagrams and hierarchical knowledge structures that facilitate the adoption of commands [60]. In this study, we used Code.org's CMC, which comprises 20 levels and diverse gaming characters like angry birds, plants, and zombies in the episodes to make coding exercises appealing and enjoyable for students. CMC aims to satisfy the fundamental framework of computer science with a variety of instructions. In each part, the number of steps required to answer the issues is specified, and the problems at each level are distinct. Student progress is recorded so that the class administrator can monitor student development. At the novice level, the player is expected to advance a few steps. However, at higher levels, the player is expected to employ fundamental programming structures such as sequences, loops, and conditions. Students may understand the logic of the algorithm, if conditions, variables, loops, and functions with the use of this teaching material [61]. In addition, CMC allows students to obtain problem-solving, logical

thinking, critical thinking, and reasoning abilities while learning the fundamentals of computer science.

According to the advice on the Code.org website, middle school students might utilize CMC. The course includes activities based on the detection and design of pathways, the orientation of an item toward a given target (right, left, turning a known number of degrees, drawing figures using a predefined path), algorithmic problem-solving, and the creation of mini games, among others.

The Code.org instructor console allows teachers to view the progress of students on the platform. The instructor console includes a number of tools for monitoring and analyzing student work as well as managing the students in a specific area. Participants utilized programming concepts such as sequences, loop structures, and conditions to solve puzzles (Table 1).

**Table 1.** CMC levels.

| Puzzle | Line Number | Content | Concept | Class |
|--------|-------------|---------|---------|-------|
| L1 | 2 | forward | sequence | |
| L2 | 3 | forward | sequence | |
| L3 | 4 | forward-direction | sequence | 1 |
| L4 | 5 | forward-direction | sequence | |
| L5 | 8 | forward-direction | sequence | |
| L6 | 2 | forward-repeat | loop | |
| L7 | 3 | forward-direction-repeat | loop | |
| L8 | 5 | forward-direction-2 repeat | loop | |
| L9 | 3 | loop in loop | loop | |
| L10 | 2 | conditional loop | while loop | 2 |
| L11 | 4 | conditional loop | while loop | |
| L12 | 5 | conditional loop (with 5-line code) | while loop | |
| L13 | 5 | conditional loop (with 5-line code) | while loop | |
| L14 | 3 | if loop- | if | |
| L15 | 5 | if loop- | if | |
| L16 | 4 | if loop- | if | |
| L17 | 4 | if loop- | if | 3 |
| L18 | 4 | else if | else if | |
| L19 | 4 | else if | else if | |
| L20 | 4 | else if | else if | |

The test was likely based on the Classic Maze Course on Code.org mentioned earlier, consisting of 20 levels. Field experts evaluated the participants' final results from this test and segregated them into three main classes based on the complexity of the programming concepts demonstrated:

Class 1: This class encapsulates participants who demonstrated basic coding skills, and have completed levels 1 to 5 of the test. These levels involve foundational coding concepts and simple problem-solving tasks.

Class 2: Participants falling into this category exhibited intermediate coding skills, having completed levels 6 to 13. These levels involve slightly more complex concepts, involving more sophisticated logic and control structures.

Class 3: This class includes participants who demonstrated upper-intermediate coding skills, having reached levels 14 to 20. These are the most challenging levels, likely incorporating complex programming tasks that require advanced problem-solving skills, understanding of data structures, or even algorithmic thinking.

These three classes form the output parameters for the machine learning classification algorithms. In the context of machine learning, output parameters (or targets) are the values that the model tries to predict. In Figure 2, exploratory data analysis of seven features is given.

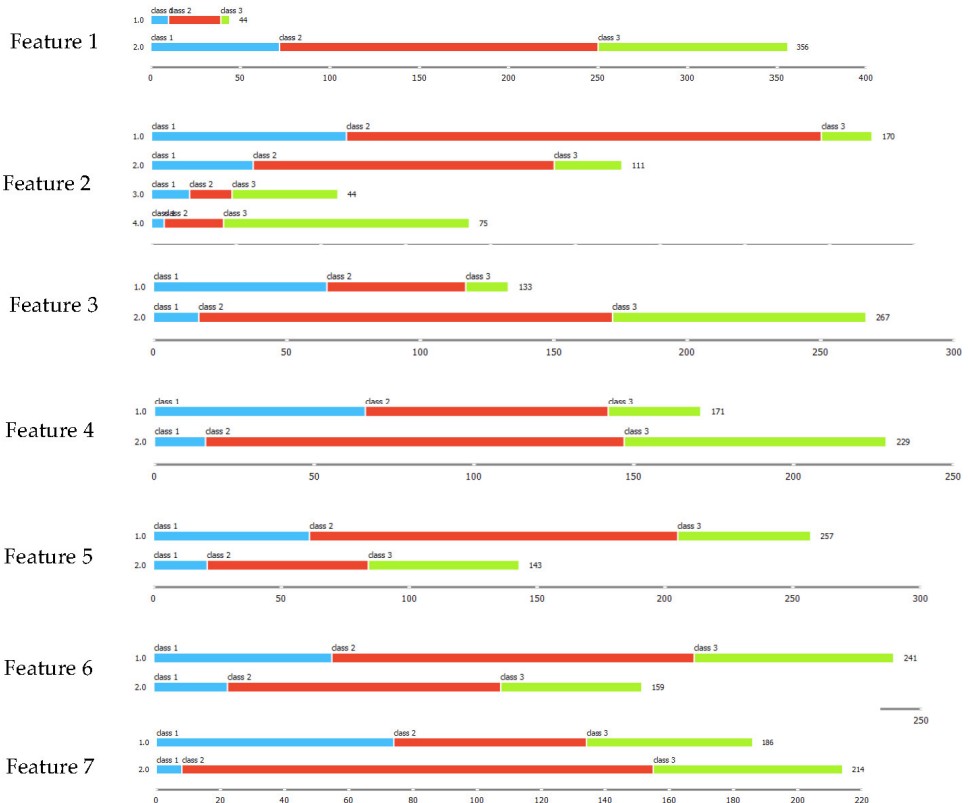

**Figure 2.** Exploratory data analysis of 7 features by different classes.

*3.4. Methodology*

Figure 3 depicts the input and output parameters of our ML classification model. The data on each participant's spatial test performance and Code.org results were collected based on 100 points and classified with ML-based classification. After a user completed each level within the required number of lines of code, they obtained a final score of 100 points.

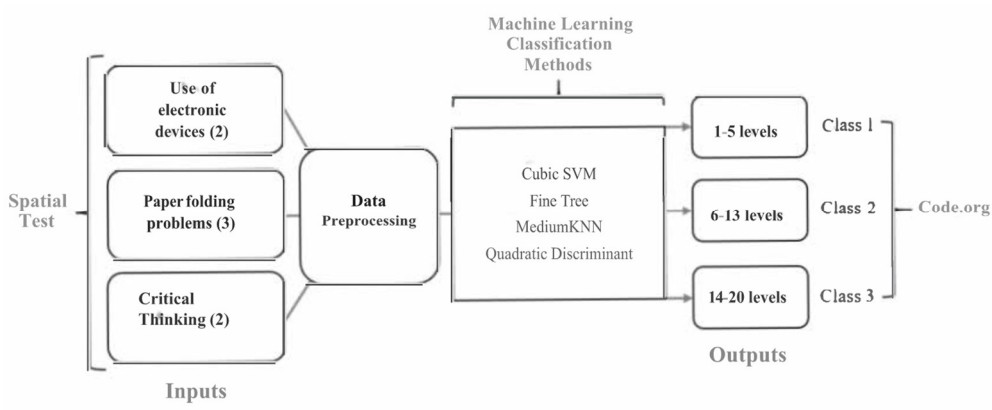

**Figure 3.** ML–based classification model used in this study.

If there were more lines of code than anticipated or missing levels in Code.org, penalty points were deducted from the final score which was utilized as the target parameter for the ML model.

Among the classification methods, cubic SVM, fine DT, medium KNN, and QD were employed owing to their superior performance.

As stated earlier, CMC puzzles were divided into three classes (Table 1) according to respective programming concepts, which were then used as an output parameter for ML model.

Cross-validation is an efficacious approach for determining a model's ability to generalize new data. In this study, we employed a stratified 5–fold cross-validation method to determine the optimal model. In this regard, we subdivided our dataset into five subgroups, or folds, with each fold containing about the same proportion of data and class labels.

To compute the performance of the four ML methods, we utilized several measures, namely, confusion matrix (CM), true positive rate (*TPR*), true negative rate (*TNR*), accuracy, precision, recall, and *F*–score.

CM is accepted as the plot of the classification of the $N \times N$ matrix.

The following equations contain formulas used to calculate *TPR*, *TNR*, accuracy, precision, recall, and *F*–score [22].

*TPR* is a measure of proportion of how many genuine positives (*P*) were discriminated against from the total number of positives detected. It is also referred to as sensitivity.

$$TPR = \frac{TP}{P} = \frac{TP}{TP + FN} \tag{1}$$

*TNR* is the ratio between the number of true negatives and the total number of negatives projected. It is also referred to as specificity.

$$TNR = \frac{TN}{N} = \frac{TN}{TN + FP} \tag{2}$$

Accuracy (*ACC*) is a model's excellence measure. The closer it is to 1, the better the model's performance.

$$ACC = \frac{TP + TN}{P + N} = \frac{TP + TN}{TP + TN + FP + FN} \tag{3}$$

Precision is defined by the number of important things selected, that is, how many of the predicted values are accurate predictions.

$$\text{Precision} = \frac{TP}{TP + FP} \tag{4}$$

In the event that precision is increasingly close to 1, our expectations become progressively more specific. Recall indicates the number of relevant items selected.

$$\text{Recall} = \frac{TP}{TP + FN} \tag{5}$$

*F*—score is the measure of accuracy. Mathematically, it is defined as follows:

$$F = 2\frac{\text{precision.recall}}{\text{precision} + \text{recall}} \tag{6}$$

Kappa value serves as an indicator of the true (corrected) accuracy of a measurement, accounting for the possibility of chance agreement. A kappa value greater than 0.4 is generally regarded as desirable.

## 4. Results and Discussion

### 4.1. Results

As stated earlier, data from 400 middle school students were used in this study. Five-fold cross validation was utilized. The feature set included answers from the students related to electronic device usage/ownership (two), paper folding problems (three), and analytical reasoning questions (two). The features ranked using the ReliefF algorithm for

classification: the spatial test questions related to paper folding, followed by electronic device usage/ownership were the most important features.

Using the whole feature set, according to the findings in Table 2, all classifiers achieved an accuracy of greater than 80%. Cubic SVM and fine DT classifiers yielded the most superior performances, with respective accuracies of 94.8% and 89.0%. Cubic SVM demonstrated the greatest classification performance in this experiment. In contrast, the accuracy of medium KNN algorithm was much lower than that of the other methods. Hyperparameters differ according to the ML algorithm. Hyperparameters are distance (Euclidean) and the number of neighbors (10) for KNN; box constraints (1), multiclass method (one vs. one), kernel scale (0.66) and kernel function (cubic) for SVM; number of splits (100) and split criterion (Gini's diversity index) for DT.

**Table 2.** Performance of different classification methods.

| Classification Method | Accuracy | Kappa | Precision | Recall | F-Score |
|---|---|---|---|---|---|
| Cubic SVM | 94.8% | 91.5% | 93.6% | 94.8% | 94.1% |
| Fine DT | 89.0% | 82.5% | 88.2% | 87.2% | 87.6% |
| Medium KNN | 80.8% | 70.0% | 79.9% | 76.4% | 77.8% |
| QD | 84.3% | 75.0% | 83.5% | 82.0% | 82.6% |

For Cubic SVM, the highest classification accuracy value is reached as the kernel scale approaches 0.66. As the kernel scale increases, classification performance decreases. A negative correlation is observed between the number of neighbors and the classification performance in KNN, wherein an increase in the number of neighbors leads to a decrease in accuracy. As the split number decreases in Fine DT, the accuracy value decreases. After a certain value, it remained constant without increasing.

An analysis was carried out on the data that were inaccurately classified by the Cubic SVM algorithm, which provided the best classification outcomes. The majority of errors occurred when students either did not finish a paper folding problem or responded inaccurately. In addition, the outliers may produce inaccurate results.

The average accuracy and recall values of the classifiers are displayed in Table 2. The accuracy and recall numbers were highest for the cubic SVM. Medium KNN had the lowest accuracy and recall values, indicating its inability to accurately identify all student classes.

Using per-class matrices such as precision and recall values, we further evaluated the performance of ML classifiers based on how successfully they could categorize certain classes. According to the total number of students assigned to each class, precision shows the percentage of accurately identified students in each class, whereas recall shows the percentage of accurately identified students among all the students in each class.

When a classification model generates predictions on data, the CM examines the model's performance and indicates how well the classification model performs. Using the CM, several model parameters, including accuracy and precision, may be computed. The receiver operating characteristic (ROC) curve is generated by plotting the false positive rate and true positive rate on the x-axis and the y-axis, respectively. In other words, the ROC curve graph is the ratio of the true positive rate to the false positive rate. The approximation of the field below the ROC curve (area under the curve, AUC) value from 0 to 1 on the graph shows that positive and negative values are properly distinguished. The CM and ROC curve of the algorithms in each dataset are depicted in Figures 4–7.

A comparison between cubic SVM CM (Figure 4) with the CMs of other algorithms reveals that each class of data is predicted with greater accuracy than the others. When examining the ROC curve for cubic SVM, the AUC was found to be 0.99. Taking into account the CM and ROC curve, it is possible to conclude that cubic SVM is superior to other approaches for this classification.

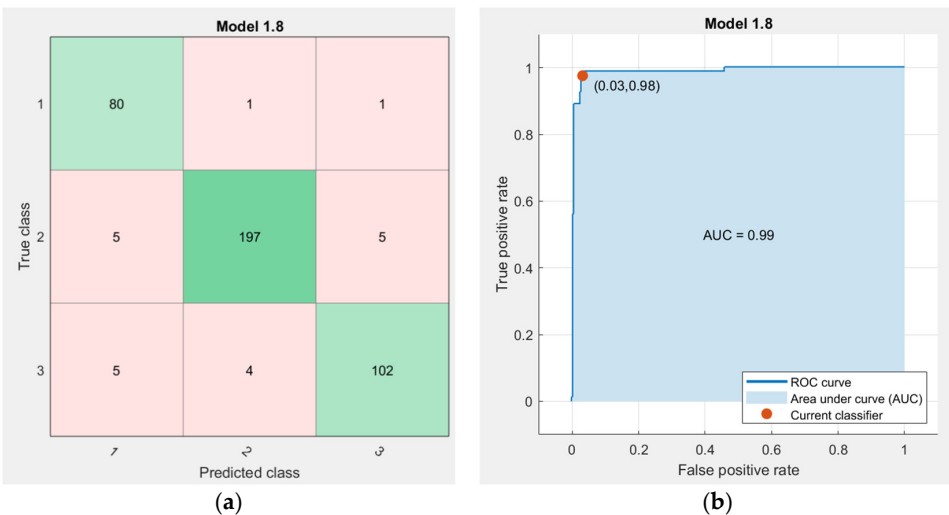

**Figure 4.** (**a**) Cubic SVM CM; (**b**) Cubic SVM ROC curve.

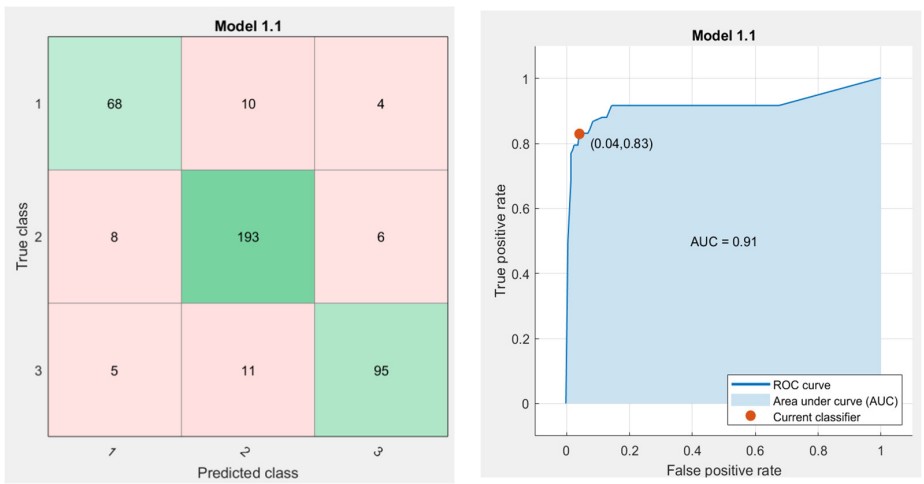

**Figure 5.** (**a**) Fine DT CM; (**b**) Fine DT ROC curve.

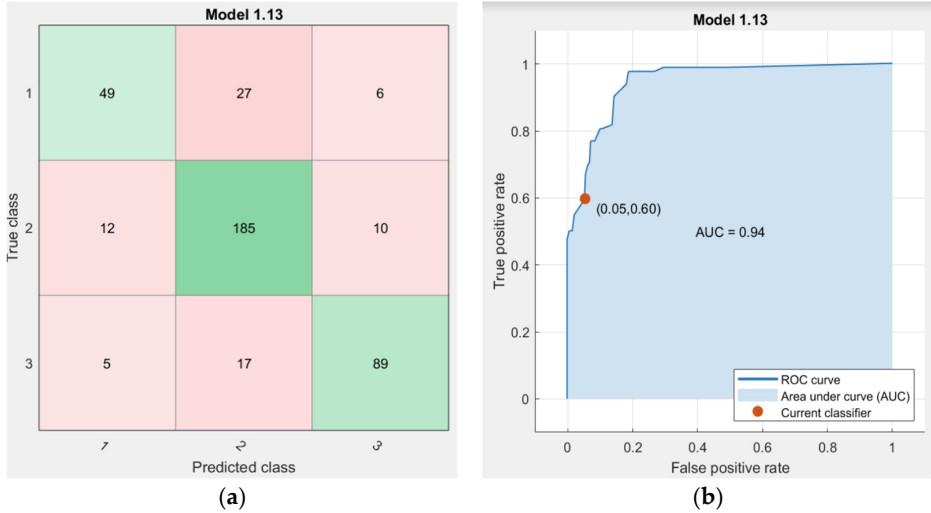

**Figure 6.** (**a**) Medium KNN CM; (**b**) Medium KNN ROC curve.

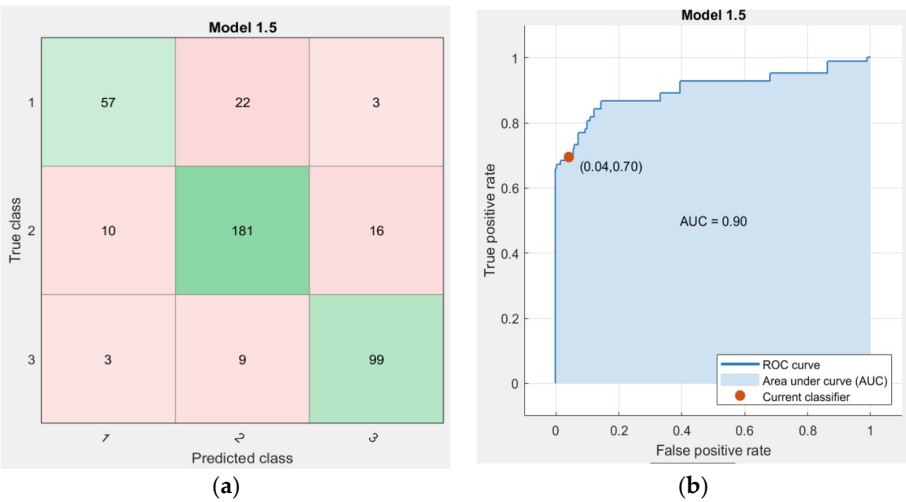

**Figure 7.** (**a**) QD CM; (**b**) QD ROC curve.

The CM and ROC curve for the fine DT algorithm are presented in Figure 5. The number of correctly estimated data in the CM is less than the cubic SVM, and the field below the ROC curve is AUC = 0.91.

The CM and ROC curve for the medium KNN are depicted in Figure 6. The number of correctly predicted data in the CM is less than the cubic SVM and fine DT. The field below the ROC curve is AUC = 0.94.

The CM and ROC curves for the QD algorithm are given in Figure 7. The number of correctly predicted data in the CM is less than the cubic SVM and fine DT but more than medium KNN. The field below the ROC curve is AUC = 0.90.

### 4.2. Discussion

Table 3 provides a summary of the results of this study and other significant classification studies of student performance. The purpose section is written in bold to show which school level of students the articles are aimed at. Considering the table, it can be said that the SVM method is more successful than other methods in classifying educational data.

**Table 3.** A summary of student performance classification techniques.

| Paper | Purpose | Algorithm | Accuracy | Data |
|---|---|---|---|---|
| [62] | To examine and evaluate the performance of school students using classification algorithms. | RF | 89.23% | 100 |
| [63] | To apply classification algorithms to the prediction of students' performance on semester-ending university exams. | NB | 70.00% | 250 |
| [64] | To investigate and evaluate the performance of university students using various classification methods. | Bayes net | 92.00% | 225 |
| [65] | To examine and evaluate the performance of high school students | SVM | 78.00% | 459 |
| [66] | To investigate and evaluate the performance of students | J48 | 97.21% | 32,593 |
| [67] | To investigate and evaluate the performance of university students | SVM | 80.00% | 340 |
| The current study | To classify the coding abilities of middle school children using spatial test and Code.org platform information. | Cubic SVM | 94.80% | 400 |

The findings of this study are consistent with the findings of other studies that have investigated the effectiveness of the SVM method for classifying educational data. For example, a study by Zafari et al. [54] found that the SVM method achieved an accuracy of 78.00% in evaluating the performance of high school students during the semester. Another study by Triayudi et al. [56] found that the SVM method achieved an accuracy of 80.00% in predicting student learning habits and steps that could be taken to improve student achievement at the university.

These findings suggest that the SVM method is a promising approach for classifying educational data. However, more research is needed to further validate the findings of this study and to investigate the effectiveness of the SVM method for classifying different types of educational data.

## 5. Conclusions

Rapid breakthroughs in AI and softwarization might substantially alter labor markets [68]. Therefore, a new class of workers equipped with programming and data analytics skills is necessary to manage it.

Data analytics and programming are essential components of enterprises in various industries. Therefore, the sooner students are exposed to a curriculum that includes coding, the sooner they will be able to profit from career opportunities at these institutions. Regarding the first research question on determining whether the coding skills of middle school students can be evaluated using a spatial test, the investigation of the relationship between the spatial test used in the evaluation and coding skills yielded favorable results. This suggests that the spatial test has the potential to be an effective instrument for evaluating and assessing the coding skills of middle school pupils. This study contributes to the increasing body of evidence supporting the use of spatial tests as a valuable assessment method for measuring coding proficiency among students of this age.

The study employed four distinct machine learning (ML) techniques to investigate the predictability of middle school students' programming abilities. Cubic Support Vector Machine (SVM) emerged as the most effective method for producing superior results. This finding addresses the second research question and suggests that the implementation of cubic SVM holds great promise for accurately predicting the programming aptitude of students at this level of education.

The findings of this study can also be utilized by educators and parents to encourage students to pursue careers in programming-related industries. If a student with no previous programming experience performs highly on the spatial test, which is successfully employed in this study to predict student programming skills, parents and instructors should be advised to encourage the student to seek a career in programming. This model, as with other prediction models, should not mean that relatively unsuccessful students cannot work in these careers. Evaluation of student performance is useful in helping teachers better support struggling students while assigning more challenging homework to high performers. In addition, the academic progress of students who complete the introductory programming course despite low performance may be monitored more closely in the future.

The implications of this study for policymakers include the development of policies to support the development of coding skills in middle school pupils. This can be accomplished by funding coding programs, training instructors, and providing students with coding resources.

The application time interval of the study's activities was constrained by participants' weekday school schedules. Due to school administration processes, participation in weekend programs was restricted. This study was conducted in a certain region and thus cannot be generalized globally. On the other hand, it gives a comprehensive investigation of a subject using a certain number of samples.

This article addresses the United Nations Sustainable Development Goals by promoting quality education and utilizing technology and innovation to identify students with coding skills, thereby contributing to the development of a competent workforce, and fostering sustainable industrialization.

**Author Contributions:** Conceptualization, A.Ç. and Ö.K.B.; methodology, Ö.K.B.; software, A.Ç. and H.K.; validation, H.K., A.Ç. and Ö.K.B.; formal analysis, A.Ç.; investigation, A.Ç.; resources, A.Ç. and H.K.; data curation, A.Ç.; writing—original draft preparation, A.Ç. and H.K.; writing—review and editing, H.K.; visualization, A.Ç.; supervision, Ö.K.B. All authors have read and agreed to the published version of the manuscript.

**Funding:** This research received no external funding.

**Institutional Review Board Statement:** Not applicable.

**Informed Consent Statement:** No private information of the subjects was used in this study.

**Data Availability Statement:** Any datasets used and/or analyzed during the current study are available from the corresponding author on reasonable request.

**Acknowledgments:** This paper is based on Ali Çetinkaya's Ph.D. dissertation. The authors would like to acknowledge the support of the Ministry of National Education Konya Provincial Directorate of Turkey.

**Conflicts of Interest:** The authors declare no conflict of interest.

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
