# Peer review of "Analysis of Machine Learning Classification Approaches for Predicting Students’ Programming Aptitude"

_sustainability, doi:10.3390/su151712917_

Round 1

Reviewer 1 Report

The paper is well structured and written. I have the comments below:

1) please cite more references for the definition of : Machine learning (ML) is a subfield of artificial intelligence (AI) and computer sci- 72 ence (IBM, 2022) that concentrates on the use of data and algorithms to simulate the way 73 people learn while continually enhancing the accuracy of these algorithms.

2. It is suggested to separate the section of research purpose and questions from the section Introduction.

3. It is suggested to highlight the key work using sub-headings for section 2. Related Work.

4.  In the section of 3. Materials and Methods, more information about the participants and their school is needed. One more, the course implementation process should be introduced as well.

5. Please also highlight the key findings or results using sub-headings for the section of Results and Discussion. It is suggested to improve the discussions of your findings by echoing the related studies.

Reviewer 2 Report

The strong part of the paper is that the authors provide quite detailed context of the literature. The presentation of the classification algorithms evaluation was also quite adequate.

However in the current form reviewer has a serious issue regarding understanding of what is the input of the classification algorithms and what are exactly the classes that are classified. And what exactly was performed as data preprocessing.

Additionaly reviewer suggest so expand initial exploratory data analysis of the features.

Several questions that should be adressed:
How many features were used as the input? 

What were the most significant features for the classification ? additional analysis of interpretability for maching learning methods is welcome.

What were the hyperparameters of the performed models tuned?

Did authors analysed the mistakes done of the models? Maybe mistakes were related to some data anomalies

Round 2

Reviewer 1 Report

1.       Can you check the format, whether they are suitable?  for example, in section of 2. Related Work , I see many In[28], In [30]…..

2.       Please improve the section of 4.2 Discussion, it needs further discussion based on your research results with echoing the related studies.

Reviewer 2 Report

I see quite adequate improvement of the paper according to the previous recommendation.

Some ideas which can be used for a futher improvement:

- exploratory data analysis (visualization) of the mentioned 7 features by different class

- analysis of the influence of the hyperparameters on the classification results
